# Marginal Zone B (MZB) Cells: Comparison of the Initial Identification of Immune Activity Leading to Dacryoadenitis and Sialadenitis in Experimental Sjögren’s Syndrome

**DOI:** 10.3390/ijms241512209

**Published:** 2023-07-30

**Authors:** Ammon B. Peck, Julian L. Ambrus

**Affiliations:** 1Department of Infectious Diseases and Immunology, College of Veterinary Medicine, University of Florida, Gainesville, FL 32610, USA; 2Division of Allergy, Immunology and Rheumatology, SUNY Buffalo School of Medicine, 875 Ellicott Street, Buffalo, NY 14203, USA; jambrus@buffalo.edu

**Keywords:** Sjögren’s syndrome, dacryoadenitis, keratoconjunctivitis, marginal zone B (MZB) cells, RNA transcriptome microarrays, C57BL/6.NOD-*Aec1Aec2* mice, B6.*Il14α* mice, type 1 interferon, signal transduction pathway

## Abstract

Although multiple mouse strains have been advanced as models for Sjögren’s syndrome (SS), which is a human systemic autoimmune disease characterized primarily as the loss of lacrimal and salivary gland functions, the C57BL/6.NOD-*Aec1Aec2* recombinant inbred (RI) mouse derived from the NOD/ShiLtJ line is considered one of the more appropriate models exhibiting virtually all the characteristics of the human disease. This mouse model, as well as other mouse models of SS, have shown that B lymphocytes are essential for the onset and development of observed clinical manifestations. Recently, studies carried out in the C57BL/6*.IL14α* transgenic mouse have provided clear evidence that the marginal zone B (MZB) cell population is directly involved in the early pathological events initiating the development of the clinical SS disease, as well as late-stage lymphomagenesis resulting in B-cell lymphomas. Since MZB cells are difficult to study in vivo and in vitro, we carried out a series of ex vivo investigations that utilize temporal global RNA transcriptomic analyses to profile differentially expressed genes exhibiting temporal upregulation during the initial onset and subsequent development of pathophysiological events within the lacrimal and salivary gland tissues per se or associated with the leukocyte cell migrations into these glands. The initial transcriptomic analyses revealed that while the upregulated gene expression profiles obtained from lacrimal and salivary glands overlap, multiple genetic differences exist between the defined activated pathways. In the current study, we present a concept suggesting that the initial pathological events differ between the two glands, yet the subsequent upregulated TLR4/TLR3 signal transduction pathway that activates the type-1 interferon signature appears to be identical in the two glands and indicates an autoimmune response against dsRNA, possibly a virus. Here, we attempt to put these findings into perspective and determine how they can impact the design of future therapeutic protocols.

## 1. Introduction

Autoimmune diseases generally arise in genetically or physiologically predisposed hosts through multi-step processes that are often initiated by unidentified environmental triggers. At the molecular and cellular levels, it is now possible to define this developmental process as an initial activation of an innate inflammatory reaction that subsequently progresses to the adaptive immune response that targets critical biological functions. These targeted functions eventually lead to cellular dysfunctions and/or cell death, eventually resulting in an overt clinical pathology [1,2,3,4,5,6]. As such, autoimmune diseases are considered to have both an early covert disease phase of inflammation, followed by a late overt disease phase of targeted organ or tissue destruction.

Unfortunately, due to this latent phase, patients generally present in the clinic only after the adaptive immune phase is active and irreversible pathology is occurring. This latter situation, therefore, necessitates the early identification of those autoimmune-associated molecular and cellular processes involved with the early covert inflammatory phase of autoimmunity as a basis to better understand the initiation of the adaptive immune response phase in order to be able to design better intervention therapies. This is especially pertinent for Sjögren’s syndrome (SS), where the time between the disease onset and eventual diagnosis ranges from 4 to 10 or more years [7,8,9]. Furthermore, the pathological profiles at the cellular and molecular levels appear to change dramatically over the course of SS disease development and onset [10].

SS is a highly debilitating systemic autoimmune disease marked primarily by leukocytic infiltrations of the salivary and lacrimal glands with a concomitant loss of exocrine secretion, leading to clinical symptoms of severe dry mouth and dry eyes [11]. Occasionally, patients develop kidney and lung disorders, central nervous system manifestations (especially mind fogginess), and even lymphomas (especially non-Hodgkin’s B cell lymphomas) [7,12,13]. Although considerable efforts have focused on defining the general pathophysiology and accompanying symptoms of the disease, more recently, a concerted effort has surfaced to better define the molecular basis for the onset of theF subsequent development of SS autoimmunity and lymphomagenesis. Much of this research has been carried out utilizing a wide variety of mouse models [14,15,16,17].

Over the past several decades, many investigators have studied multiple mouse models that exhibit SS-like pathologies as a means to define a variety of disease parameters that presumably affect cellular and molecular events that are responsible for the development and onset of SS in humans [18,19,20,21,22,23]. In general, these studies have defined this SS-like disease development as the following: (a) an early aberrant proteolytic enzyme activity that is most likely associated with acinar cell apoptosis [18,19], (b) a progressive loss of saliva and tear flow rates accompanied by a general increase in protein content due in part to the loss of fluid secretions [20], (c) a decline in amylase and carbonic anhydrase activities [24], (d) the appearance of autoantibodies [21,22], and eventually (e) the loss of acinar tissue [23]. These disease manifestations occur concomitantly with increasing glandular lymphocytic infiltrates primarily comprising adaptive immune cell types, especially T lymphocytes [25,26,27,28]. In addition, recent studies have shown a direct correlation between an upregulated expression of interleukin IL14α and late-stage B-cell lymphomagenesis [29,30,31], completing the SS phenotype in these mouse models. At the same time, global temporal transcriptome studies verified these reported immune–pathophysiological temporal changes at the gene level [32]. From these observations, a simplified disease profile is presented in Table 1 and makes a general comparison of SS parameters between human SS and the mouse SS-like disease, and shows why our current murine models are highly appropriate for the study of the human disease.

Although there have been a multitude of murine models that have impacted the identification of a large array of factors defining the SS-like disease in mice [33], there are four genetically manipulated lines that have had a significant impact, including the NOD/Shi (or NOD/ShiLtJ) mouse [34], the C57BL/6.NOD-*Aec1Aec2* recombinant inbred (RI) line [19], the *TSP*^-/-^ gene knock-out (KO) mouse [35], and the *IL14α*^-/-^ gene knock-in (KI) mouse [29]. The NOD/Shi mouse represents a secondary SS phenotype and the C57BL/6.NOD-*Aec1Aec2* mouse represents a primary SS phenotype, while the *TSP*^-/-^ and *IL14α*^-/-^ gene KO and KI lines represent unique models for dacryoadenitis and B-cell lymphomagenesis, respectively. Interestingly, earlier studies, specifically with *Baff* [36,37] and *Igµ* [38] gene KO lines, revealed that the inactivation of the B-cell receptor in various mouse models prevents the development of an SS-like disease. Recent studies with B6.*Il14α* transgenic (TG) mice indicated that the elimination of marginal zone B (MZB) cells, or the blocking of lymphotoxin activity that is required for MZB cell ontogeny in marginal zones (MZ), prevents the development of an SS-like disease, including subsequent lymphomagenesis [39,40].

Over the past few years, the interest in studying the role of MZB cells in SS has increased significantly. Multiple recent publications resulting from in-depth studies, mostly from using the above-mentioned mouse models of SS-like disease, have appeared; thus, the reader is directed to several excellent research papers describing how MZB cells might influence the development and subsequent regulation of SS [26,40,41,42,43,44]. In the present report, we attempt to put our recent studies into perspective by comparing the SS-like disease profiles of the lacrimal versus the salivary glands based on differential upregulations of gene transcription and gene sets that define cellular signal transduction pathways in the early innate phase of an SS-like disease.

## 2. Results

### 2.1. Time-Dependent Leukocytic Infiltrations of the Lacrimal Gland in the C57BL/6.NOD-Aec1Aec2 Mouse Model of Primary Sjögren’s Syndrome

As presented in Figure 1, histological evaluations of the lacrimal glands of C57BL/6.NOD-Aec1Aec2 mice provide clear evidence for an age-dependent progressive cellular autoimmune attack within this glandular tissue that can be visualized as early as 8–10 weeks of age. This temporal histological evaluation also reveals the periductal origin and nature in the infiltrating cells. Over time, these small periductal infiltrations develop into the characteristic lymphocytic foci (LF) that are common to both human SS and this mouse model of primary SS. These histological profiles and the temporal appearance of infiltrating cells in lacrimal glands are also characteristic of LF development in salivary glands [45]. While C57BL/6.NOD-*Aec1Aec2* mice do not show any lymphomagenesis development for up to 9–10 months of age, the *IL14α*^-/-^ gene KI mice commonly develop lymphomas by 14–18 months of age [29].

### 2.2. Time-Dependent Pathological Events and Their Appearance in the Lacrimal and Salivary Glands in the C57BL/6.NOD-Aec1Aec2 Mouse Model of Primary Sjögren’s Syndrome

The development and onset of SS represents a methodical activation, maturation, and modification of multiple immune–pathophysiological biological processes that appear and are subsequently modified as the disease progresses through multiple stages [10,47,48]. Many of these changes have been defined through studies of both human SS and various mouse models of SS-like disease. Although the model remains in constant flux, at least four phases are clearly definable, including (i) an initial innate immune response, (ii) a transitional immune response to an adaptive response, (iii) a mature adaptive immune response, and (iv) severe loss of glandular tissue and function with a potential for evolution to a lymphomagenesis [13,49,50,51]. These phases can be defined by sets of distinct biological actions, several of which are presented in Figure 2.

### 2.3. Comparison of the Notch2 Receptor and Signal Transduction Pathway Gene Profiles Expressed in Lacrimal versus Salivary Gland MZB Cells during the Early Innate Immune Phase of Sjögren’s Syndrome in C57BL/6.NOD-Aec1Aec2 Mice

Although early studies using the NOD/ShiLtJ mouse model of SS [52] indicated an important role for MZB cells in the development and onset of SS-like pathologies, this fact was not identified until studies with the *BAFF^-/-^* [53], *TSP^-/-^* [54], and B6.*Il14α* [40] models were published. Nevertheless, the recent studies of Shen et al. [40] have provided definitive proof in studies that eliminate the MZB cell population per se and studies that block the lymphotoxin-α activity required for MZB cell ontogeny in MZs [39].

MZB cells differentiate from transitional type-1 (T1) B cells under the influence of low affinity B-cell receptor (BCR) signals and transcription factors, especially Notch-2 [55,56,57,58]. The activation of MZB cells involves the stimulation of their BCR complex by delta-like ligands or Hedgehog ligands that, in turn, activate the Notch2 signal transduction pathway that, in the absence of Notch1, Notch3, or Notch4 co-activation, is unique to MZB cells. As shown in Figure 3, a common Notch2 signal transduction pathway appears to be present/activated in both the lacrimal and salivary glands despite the fact that each can use different Jag (Jag1 versus Jag2) and Dtx (Dtx4 versus Dtx1) molecules. However, the Notch2 signal transduction profile(s) exhibited in the salivary glands suggest(s) a possible second pathway using the receptor Mfng and several unique signaling molecules.

### 2.4. The Upregulated Gene Expression Profiles for the Interferon Type-1 Signal Transduction Pathways Expressed in Lacrimal and Salivary Glands of C57BL/6.NOD-Aec1Aec2 Mice Are Not Only Similar, but Expressed during Identical Time Frames

While Figure 2 presents a simplified picture of multiple pathophysiological and cellular events identified as important temporal events appearing during the onset of SS-like disease in various mouse models, especially NOD/ShiLtJ, C57BL/6.NOD-*Aec1Aec2*, and B6.*Il14α* KI mice, one characteristic of SS-like disease is an early appearance of alarmins and lymphotoxin, followed by an interferon type 1 (IFN-1) signature with multiple upregulated interferome genes. MZB cells are known to produce lymphotoxin [39,59], which can be involved with the formation of local lymphoid follicles as well as a direct cellular cytotoxicity [60,61]. MZB cells are also known to activate a type 1 interferon response via the stimulation of their Toll-like receptors (TLRs) [62], and we have previously shown that C57BL/6.NOD-*Aec1Aec2* mice exhibit upregulated expressions of *Anxa* (annexin), *Tlr3*, and *Tlr4* genes during the early innate response [7,63,64]. Our transcriptome data also demonstrate that the genes comprising the complete interactive Tlr3/Tlr4 signal transduction pathways are co-upregulated, as presented in Figure 4, and point to a strong interferon signature activation. This upregulated signaling pathway is found in both the salivary and lacrimal gland tissues of C57BL/6.NOD-*Aec1Aec2* mice as early as 8–12 weeks of age, with an upregulated *Tlr3* expression that is still present at 20 weeks of age (data not presented). Although this profile is consistent with an immune response towards dsRNA, it remains unclear whether this response is towards a virus.

## 3. Discussion

Over the past several decades, our research programs have both independently and in collaboration focused on deciphering the intricate pathophysiological and cellular events underlying human SS by utilizing multiple murine models exhibiting SS-like disease. Using a genetic and molecular approach, we identified multiple biological changes that occur within the targeted tissues during the onset, development, and destructive autoimmune stages of the disease, as well as the onset of lymphomagenesis. Many of these disease-associated genetic and pathological events were documented in detail, but one important and fascinating aspect that remains to be clarified is whether the autoimmune responses that are observed within the lacrimal and salivary glands, as well as other targeted tissues such as the lung and kidney, are identical, similar, or different. Non-identical immune signatures would most likely require multiple approaches for effective clinical treatments, especially in light of the multiple disease phenotypes that are present in SS patients. The evidence from animal models demonstrate that damage to various organs may occur via different immune mechanisms [59,65].

Interestingly, to study and analyze the human SS disease, multiple animal models were identified or genetically generated that mimic different aspects of the human disease. Most animal models demonstrate only a partial disease phenotype. Nevertheless, despite the broad spectrum of the human SS disease, several mouse models were identified that produce what appears to be pathologies or partial pathologies that are quite similar to the human disease. The degree to which the murine SS-like disease corresponds to human SS should become evident when clinical trials are performed with immune-modifying drugs developed based on data from murine systems.

For the past several decades, the major focus in studying SS in both humans and animal models has been to identify the role of autoantibodies, antigen-presenting cells, T lymphocyte subsets, and NK and innate lymphoid (ILCs) cells to define the characteristics of the autoimmune response. In 1998, Robinson et al. [38] published studies showing that knocking out the *Igµ* gene in NOD mice totally prevented the development of an SS-like disease. Additional studies designed to interfere with B-cell functions have supported the importance of B cells in multiple aspects of SS pathology and the onset of SS disease [66]. While considerable attention has been placed on the role of B cells in antibody production, recent studies in several mouse models have provided evidence that marginal zone B cells that are capable of regulating multiple immune cell functions play a critical role in the development of anti-glandular autoimmunity and subsequent systemic clinical SS-like disease [40,42,44]. It is not surprising, then, that MZB cells have been identified within the salivary glands of patients with salivary gland disease where they secrete cytokines that are cytotoxic to salivary gland cells [39,41].

Our studies in the C57BL/6.NOD-*Aec1Aec2* mouse model revealed that, except for the occasional sentinel leukocytes that normally traverse through healthy lacrimal and salivary gland tissues, an MZB cell population appears to be the first pathogenic cells to emigrate into the lacrimal and salivary glands, and they emigrate concomitantly just after the upregulated gene expressions of the annexin genes (*Anxa1-6* and *Anxa11*) that are classified as alarmins [64]. This feature was determined by both histology (Figure 1) and gene expressions (Figure 4). This progressive nature of the SS-like disease is further supported by the concomitant upregulated expressions of the *Cxcl13* chemokine and its receptor *Cxcr5* in the lacrimal and salivary glands that are at their maximum around 16 weeks of age [42,43].

MZB cells are a unique subpopulation of bone-marrow-derived B cells characterized by a limited expression of immunoglobulin variable region genes that produce predominantly IgM antibodies, many of which are self-reactive [55,56]. They differentiate from transitional type-1 (T1) B cells under the influence of low-affinity B-cell receptor (BCR) signals and transcription factors, especially Notch-2 [55]. MZB cells are strategically located within mucosal surfaces, function as innate and/or transitional cells that are capable of rapid responses to both T-cell-independent and T-cell-dependent antigens, and can help to regulate the activation of subsequent adaptive immune responses by T and B2 lymphocytes in association with monocytic and neutrophilic antigen-presenting cells (APCs) (reviewed in [56]). They are enriched within splenic MZs, retained there by interactions between MZ integrin and MZB cell integrin receptors, especially MADCAM1 (mucosal cell adhesion molecule-1), LFA1 (lymphocyte function-association-1), ICAM (intercellular adhesion molecule-1), VLA-4 (very late antigen-4), and SIP1 (sphingosine-1-phosphate) [25]. Functionally, MZB cells respond rapidly to Toll-like receptor (TLR) signaling, especially TLR2, 4, and 9 [67]. In C57BL/6.NOD-*Aec1Aec2* mice, our molecular gene analyses indicate a strong response by Tlr3 in the SS-like disease (Figure 4) that appears to interact with a concomitantly expressed Tlr4 response to upregulate the gene expressions of an interferon signature, multiple cytokines, and Mapk14. Importantly, MZB cells are known to be important in the cellular responses to pathogens, especially viruses [68,69].

The indication of an activated Tlr3 receptor that is stimulated by a dsRNA entity, possibly a virus, is quite intriguing in light of the earlier studies carried out in NOD-*scid* mice that lack both T and B cells [6]. These *scid* mice undergo a spontaneous Fas:FasL-dependent apoptosis of glandular secretory epithelial cells and show a loss of acinar cells accompanied by an increased level of the ductal cell component despite having a lack of lymphocytes. Consistent with this finding is the fact that there are increased levels of cysteine proteases in the submandibular glands of both NOD-*scid* and the parental NOD/ShiLtJ mice. Although the presence of activated lymphocytes appears to be necessary in NOD/ShiLtJ mice to develop an overt autoimmunity, the observation of high levels of apoptosis and aberrant protein expressions in the submandibular glands of NOD-*scid* mice, even in the absence of an immune response, suggests that genetic alterations in glandular homeostasis involving the cell death program contributes to disease progression or even an initial trigger of SS autoimmunity. We assume that this scenario is true for lacrimal glands, but this has yet to be examined. Nevertheless, again, the loss of acinar tissue in the absence of lymphocytes raises the question of whether SS/SS-like disease has a viral etiology, initiated from damaged mitochondria, or a yet an unknown stimulation. Hopefully, the use of genomic analyses with murine models will eventually identify the very first cellular event(s) that initiate lacrimal and salivary gland SS pathology in order to develop therapies that are effective in a pre-disease phase.

## 4. Conclusions, Summary, and Contributions to the Field

In the current review, we present a simple perspective based primarily on our recently published data that focuses on the onset and temporal development of SS-like disease in mouse models that exhibit an SS-like pathology [14,16,17]. The general SS-like disease profiles defined by these models are being further investigated by recent global temporal gene transcriptome studies carried out primarily using the C57BL/6.NOD-*Aec1Aec2* and B6-*Il14α* models of primary SS [26,40,42,43,64,70]. While this perspective format does not contain a Materials and Methods section, we direct the reader to [40,42,43] for details and the Data Availability Statement below for the gene data. Interestingly, these analyses have revealed that the progressive immune-pathophysiologic SS-like disease that is present in the lacrimal and salivary glands exhibit very similar, but not identical, gene expression profiles [42,64]. Whether these observed differences result from inherent differences in the glandular tissue, autoantigen expressions, chemokine signaling, the responding MZB cell populations, or an undefined mechanism remains an open question. However, our overall gene expression data are beginning to suggest that these observed differences are more prevalent during the early innate phase than later during the transition and adaptive phases, which is a concept supported by comparing the Notch2 with the interferon gene expression profiles presented herein.

The fact that the Tlr4/Tlr3 signal transduction pathways of both lacrimal and salivary gland tissues appear to be identical in the time of upregulated gene expression and the set of pathway genes provides support for the involvement of at least one common autoantigen targeted in the two glands. Whether this apparent response is against a dsRNA virus, mitochondria, or apoptotic cellular debris represents a critical issue that will require considerable investigation. For years, multiple viruses have been suggested to underlie SS in humans, yet no virus has been definitively identified to date despite the importance of identifying the causative element(s) for initiating an autoimmune attack against the exocrine tissues. At the same time, RNA and DNA elements from mitochondria, which could be damaged by multiple different mechanisms, have been implicated in autoimmune diseases, such as SLE and SS [71,72,73]. In addition, dysfunctional mitochondria have been identified in many patients with SS [74,75,76,77,78], consistent with the wide-range of pathologies observed in SS patients. Thus, we continue to define the molecular and cellular aspects of SS using both mouse models of SS-like disease and patients with SS to find an answer. The underlying cause (or causes) for SS may vary in different patients, but most likely requires specific common key elements to initiate and maintain autoimmunity. The identification and targeting of these elements should further increase therapeutic options, but functional targets must first be identified and then shown to be effective targets, especially early targets such as alarmins, dsRNA entities, metabolic products, and pathophysiologically critical pathways. Concomitant testing in both models and patients could facilitate the development of new drugs for unique targets.

## Figures and Tables

**Figure 1 ijms-24-12209-f001:**
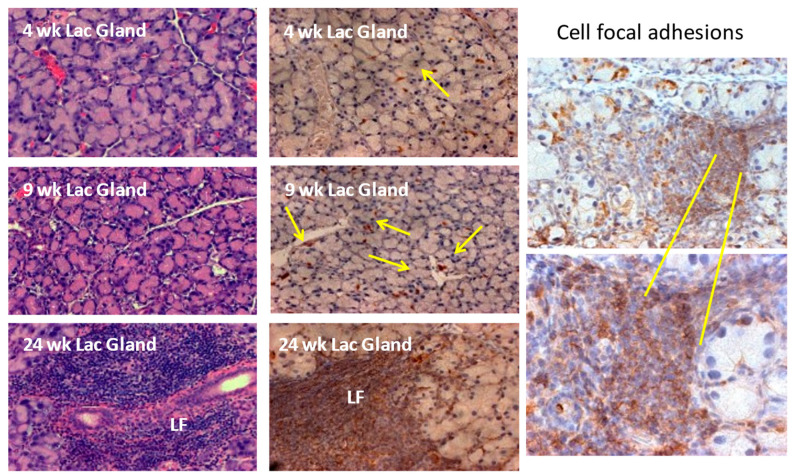
Histological photomicrographs depicting the progressive development of lymphocytic infiltrations within the lacrimal glands of C57BL/6.NOD-*Aec1Aec2* mice. Histological sections were prepared from freshly excised extra-orbital lacrimal glands euthanized at the designated ages and stained with either H&E (left panels) or rat anti-mouse anti-phosphorylated paxillin antibody (center and right panels). Paxillin is specific for focal adhesions associated with cell migration, thus identifying the time-dependent increased levels of infiltrating cells (yellow arrows). Magnification = 200X. This figure is reproduced from [46].

**Figure 2 ijms-24-12209-f002:**
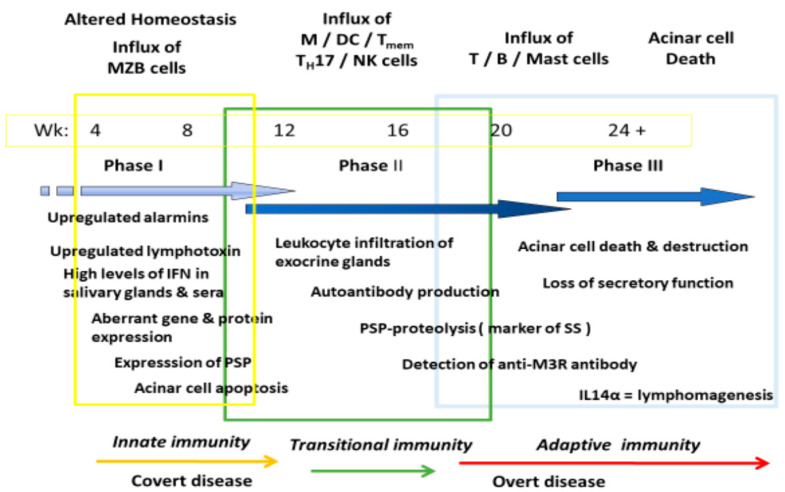
An updated general overview of major temporal changes identified in the progressive development of Sjögren’s syndrome pathology. Based on physiological and histological changes and the various time points for the appearance of specific immune cell populations within the lacrimal and salivary glands, it is possible to divide the overall disease into three main compartments, designated by the different colored boxes and arrows. Importantly, these general disease profiles occur at the same approximate time periods within the lacrimal and salivary glands. In addition, the onset of lymphomagenesis, primarily B-cell lymphomas, appears to be a function of interleukin IL14α (alpha-taxilin) expression, in that the *IL14α* gene in C57BL/6.NOD-*Aec1Aec2* mice, which is upregulated in both lacrimal and salivary glands during the early innate immunity phase, is quickly downregulated following the transition to the adaptive immune phase (data not presented). These data strongly suggest that lymphomagenesis in SS requires continued IL14α stimulation. It should also be noted that many of these physiological changes are highly transitional across phases once activated.

**Figure 3 ijms-24-12209-f003:**
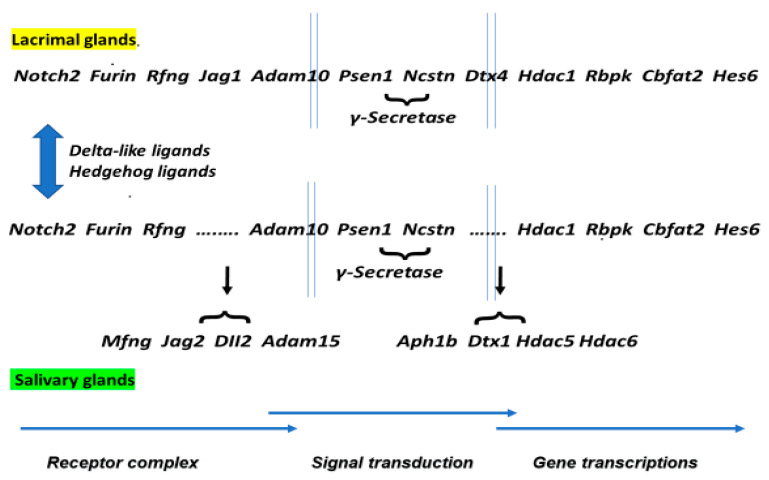
A comparison of the activated Notch2 receptor signal transduction pathways presumed to be present in marginal zone B (MZB) cells migrating into lacrimal (top pathway) and salivary glands (bottom pathways) during the innate phase of early Sjögren’s syndrome based on gene upregulated expressions. The arrows indicate how the gene sequences at *Jag1* and *Dtx4* withing the lacrimal gland are altered in the salivary gland sequence. Although the upregulated gene profile(s) observed for the MZB cells present in the salivary glands may suggest that this cell population recognizes a broader set of autoantigens than in the lacrimal gland, it remains unknown what antigens are being targeted at this early disease phase. Nevertheless, gene profiles in the two glands at this early time point appear to overlap significantly, suggesting commonalities between the immune activities developing with the two glands.

**Figure 4 ijms-24-12209-f004:**
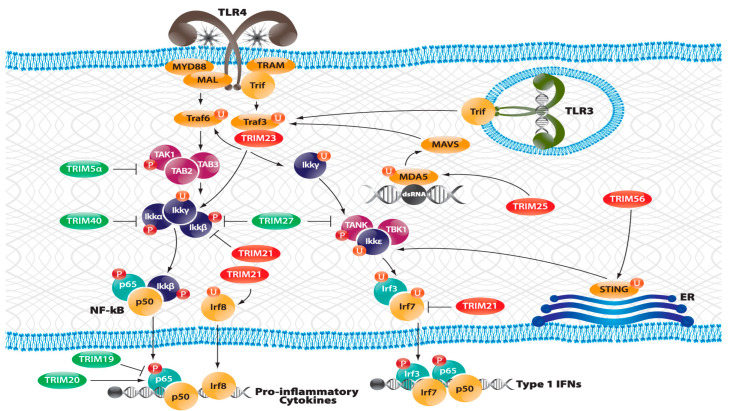
Our current gene expression profile(s) that appear(s) during the innate and early transition phases of Sjögren’s syndrome in the lacrimal and salivary glands of C57BL/6.NOD-*Aec1Aec2* mice strongly suggest a response to dsRNA, possibly against a virus or damaged mitochondria. The Toll-like receptor 3 (TLR3) is known to bind double-stranded (ds) RNA, a common entity in various families of viruses, mitochondria, and apoptotic cells. Critical for binding viruses are proteins RIG-1 (*Ddx58*), MDA-5 (*Ifih1*), Mavs (*Ips1*), and STING (*Tmem173*), the latter of which is a cytosolic pattern recognition receptor (PPR) that binds nucleic acids and transmits signals that activate type 1 interferon (IFN-1). These entities interact with the TLR4 signal transduction pathways that, in turn, lead to upregulated pro-inflammatory and interferon gene transcriptions. In addition, the TLR3 and Sting pathways are also regulated by the interferon response factors (IRFs), especially TRIM molecules, several of which enhance anti-viral pathways and show upregulated expression (shown in red). This signal transduction scheme exhibits a concomitant upregulated gene expression profile between 8–12 weeks of age in both the lacrimal and salivary glands before exhibiting decreased expressions concomitant with the appearance of the transitional and the adaptive immune response phases, which appear to activate IFNγ.

**Table 1 ijms-24-12209-t001:** A general comparison of pathological characteristics defining human SS disease with the corresponding characteristics defining murine SS-like disease. Even though SS in humans and SS-like disease in mouse models express multiple disease phenotypes, the two diseases show overall phenotypes that strongly overlap. The bracketed data are used to indicate that the major disease parameters defining SS in humans and SS-like disease in mice are themselves not universally expressed clinically, but are known to present clinically as different pathological phenotypes.

	Sjogren’s Syndrome	
Pathological Characteristics	Humans	Mice
Dacryoadentitis/Meibomian Gland Disease	(Yes)	(Yes)
Sialadenitis	Yes	Yes
Decreased tear flow rates	Yes	Variable
Keratoconjunctivitis sicca (KCS)	Yes	Variable
Ocular epithelium dessication	Yes	Yes
Decreased break-up time	Yes	(?)
Altered proteins in tears	(Yes)	Yes
Decreased Lysozyme & Lactoferrin activity	Yes	(?)
Decreased saliva flow rates	Yes	Yes
Stomatitis sicca	Yes	Yes
Altered proteins in saliva	Yes	Yes
Decreased Amylase & EGF activity	Yes	Yes
Inflammatory cytokine/chemokine production	Yes	Yes
Autoantibodies & Theumatoid Factor	Yes	Variable
Anti-Ro/SS-A, Anti-La/SS-B, Anti-DNA (ANAs), Anti-α-fodrin, Anti-β-adrenergic receptor, Anti-type-3 muscarinic ACh receptor		
Lymphomagenesis	Yes	Variable

## Data Availability

Gene Expression Omnibus, accession numbers GSE15640 and GSE36378.

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
