# Peer review of "Marginal Zone B (MZB) Cells: Comparison of the Initial Identification of Immune Activity Leading to Dacryoadenitis and Sialadenitis in Experimental Sjögren’s Syndrome"

_ijms, 2023, doi:10.3390/ijms241512209_

Round 1

Reviewer 1 Report (New Reviewer)

Manuscript ijms-2470620 by Peck et al. describes the pathophysiological and cellular events of human SS using various transgenic mouse models, and the gene expression profiling for impacted transduction pathways in MZB cells in lacrimal and salivary glands. Understanding of disease related immune responses would benefit the design of therapeutic protocol in future.

The manuscript is a well written perspective article and contains substantial amount of information for the SS field, to include contents describing pathophysiological and temporal changes over the course of SS development - innate immunity, transitional immunity, and adaptive immunity, and summarizing upregulated gene expression profiling in Notch2 receptor and INF-1 signal transduction pathways in MZB cells, which could open a new window for drug discovery. 

I only have minor comments:

In section 2.2, the title is the same as that of section 2.1

in Table 1, what is the difference between "Yes" and "Yes" in brackets

Author Response

Manuscript IJMS-2470620

Response to Reviewer 1 & Reviewer 2

We thank both Reviewer 1 and Reviewer 2 for taking time to review manuscript ijms-2470620 that was submitted to IJMS as a perspective report comparing the early pathological events occurring in the lacrimal glands versus the salivary glands primarily of the SS-susceptible C57BL/6.NOD-Aec1Aec2 mouse model.  Although several studies have been published describing both early and later events in the lacrimal glands and in the salivary glands of this mouse model, we have not directly compared these two data sets per se.  Thus, the current manuscript.

Both Reviewer 1 and Reviewer 2 identified a few weaknesses that needed to be addressed.  At this time, we have addressed these points, plus a few additional ones,  and feel that addressing these issues have greatly improved the overall paper. More specifically:

  1. The Title of the manuscript has been changed significantly to better describe the purpose of the overall paper.
  2. We have rewritten the title for Section 2.2 of the Results section.
  3. We have modified the figure legend for Figure 4.
  4. While a perspective paper does not require a methods section, we have included a statement in the Conclusion/Summary section mentioning where the data are available for viewing … in addition to the Data Availability Statement.
  5. We have rewritten much of Section 4 (Conclusions/Summary) including additional thoughts on potential future therapies.
  6. A few miscellaneous changes have been made that we believe better explains some of the data and/or figures.

All changes appear in red print for easy access.

Again, we wish to thank the Reviewers for their excellent insights and suggestions for improving the overall manuscript.

Manuscript IJMS-2470620

Response to Reviewer 1 & Reviewer 2

We thank both Reviewer 1 and Reviewer 2 for taking time to review manuscript ijms-2470620 that was submitted to IJMS as a perspective report comparing the early pathological events occurring in the lacrimal glands versus the salivary glands primarily of the SS-susceptible C57BL/6.NOD-Aec1Aec2 mouse model.  Although several studies have been published describing both early and later events in the lacrimal glands and in the salivary glands of this mouse model, we have not directly compared these two data sets per se.  Thus, the current manuscript.

Both Reviewer 1 and Reviewer 2 identified a few weaknesses that needed to be addressed.  At this time, we have addressed these points, plus a few additional ones,  and feel that addressing these issues have greatly improved the overall paper. More specifically:

  1. The Title of the manuscript has been changed significantly to better describe the purpose of the overall paper.
  2. We have rewritten the title for Section 2.2 of the Results section.
  3. We have modified the figure legend for Figure 4.
  4. While a perspective paper does not require a methods section, we have included a statement in the Conclusion/Summary section mentioning where the data are available for viewing … in addition to the Data Availability Statement.
  5. We have rewritten much of Section 4 (Conclusions/Summary) including additional thoughts on potential future therapies.
  6. A few miscellaneous changes have been made that we believe better explains some of the data and/or figures.

All changes appear in red print for easy access.

Again, we wish to thank the Reviewers for their excellent insights and suggestions for improving the overall manuscript.

Manuscript IJMS-2470620

Response to Reviewer 1 & Reviewer 2

We thank both Reviewer 1 and Reviewer 2 for taking time to review manuscript ijms-2470620 that was submitted to IJMS as a perspective report comparing the early pathological events occurring in the lacrimal glands versus the salivary glands primarily of the SS-susceptible C57BL/6.NOD-Aec1Aec2 mouse model.  Although several studies have been published describing both early and later events in the lacrimal glands and in the salivary glands of this mouse model, we have not directly compared these two data sets per se.  Thus, the current manuscript.

Both Reviewer 1 and Reviewer 2 identified a few weaknesses that needed to be addressed.  At this time, we have addressed these points, plus a few additional ones,  and feel that addressing these issues have greatly improved the overall paper. More specifically:

  1. The Title of the manuscript has been changed significantly to better describe the purpose of the overall paper.
  2. We have rewritten the title for Section 2.2 of the Results section.
  3. We have modified the figure legend for Figure 4.
  4. While a perspective paper does not require a methods section, we have included a statement in the Conclusion/Summary section mentioning where the data are available for viewing … in addition to the Data Availability Statement.
  5. We have rewritten much of Section 4 (Conclusions/Summary) including additional thoughts on potential future therapies.
  6. A few miscellaneous changes have been made that we believe better explains some of the data and/or figures.

All changes appear in red print for easy access.

Again, we wish to thank the Reviewers for their excellent insights and suggestions for improving the overall manuscript.

Manuscript IJMS-2470620

Response to Reviewer 1 & Reviewer 2

We thank both Reviewer 1 and Reviewer 2 for taking time to review manuscript ijms-2470620 that was submitted to IJMS as a perspective report comparing the early pathological events occurring in the lacrimal glands versus the salivary glands primarily of the SS-susceptible C57BL/6.NOD-Aec1Aec2 mouse model.  Although several studies have been published describing both early and later events in the lacrimal glands and in the salivary glands of this mouse model, we have not directly compared these two data sets per se.  Thus, the current manuscript.

Both Reviewer 1 and Reviewer 2 identified a few weaknesses that needed to be addressed.  At this time, we have addressed these points, plus a few additional ones,  and feel that addressing these issues have greatly improved the overall paper. More specifically:

  1. The Title of the manuscript has been changed significantly to better describe the purpose of the overall paper.
  2. We have rewritten the title for Section 2.2 of the Results section.
  3. We have modified the figure legend for Figure 4.
  4. While a perspective paper does not require a methods section, we have included a statement in the Conclusion/Summary section mentioning where the data are available for viewing … in addition to the Data Availability Statement.
  5. We have rewritten much of Section 4 (Conclusions/Summary) including additional thoughts on potential future therapies.
  6. A few miscellaneous changes have been made that we believe better explains some of the data and/or figures.

All changes appear in red print for easy access.

Again, we wish to thank the Reviewers for their excellent insights and suggestions for improving the overall manuscript.

Manuscript IJMS-2470620

Response to Reviewer 1 & Reviewer 2

We thank both Reviewer 1 and Reviewer 2 for taking time to review manuscript ijms-2470620 that was submitted to IJMS as a perspective report comparing the early pathological events occurring in the lacrimal glands versus the salivary glands primarily of the SS-susceptible C57BL/6.NOD-Aec1Aec2 mouse model.  Although several studies have been published describing both early and later events in the lacrimal glands and in the salivary glands of this mouse model, we have not directly compared these two data sets per se.  Thus, the current manuscript.

Both Reviewer 1 and Reviewer 2 identified a few weaknesses that needed to be addressed.  At this time, we have addressed these points, plus a few additional ones,  and feel that addressing these issues have greatly improved the overall paper. More specifically:

  1. The Title of the manuscript has been changed significantly to better describe the purpose of the overall paper.
  2. We have rewritten the title for Section 2.2 of the Results section.
  3. We have modified the figure legend for Figure 4.
  4. While a perspective paper does not require a methods section, we have included a statement in the Conclusion/Summary section mentioning where the data are available for viewing … in addition to the Data Availability Statement.
  5. We have rewritten much of Section 4 (Conclusions/Summary) including additional thoughts on potential future therapies.
  6. A few miscellaneous changes have been made that we believe better explains some of the data and/or figures.

All changes appear in red print for easy access.

Again, we wish to thank the Reviewers for their excellent insights and suggestions for improving the overall manuscript.

Manuscript IJMS-2470620

Response to Reviewer 1 & Reviewer 2

We thank both Reviewer 1 and Reviewer 2 for taking time to review manuscript ijms-2470620 that was submitted to IJMS as a perspective report comparing the early pathological events occurring in the lacrimal glands versus the salivary glands primarily of the SS-susceptible C57BL/6.NOD-Aec1Aec2 mouse model.  Although several studies have been published describing both early and later events in the lacrimal glands and in the salivary glands of this mouse model, we have not directly compared these two data sets per se.  Thus, the current manuscript.

Both Reviewer 1 and Reviewer 2 identified a few weaknesses that needed to be addressed.  At this time, we have addressed these points, plus a few additional ones,  and feel that addressing these issues have greatly improved the overall paper. More specifically:

  1. The Title of the manuscript has been changed significantly to better describe the purpose of the overall paper.
  2. We have rewritten the title for Section 2.2 of the Results section.
  3. We have modified the figure legend for Figure 4.
  4. While a perspective paper does not require a methods section, we have included a statement in the Conclusion/Summary section mentioning where the data are available for viewing … in addition to the Data Availability Statement.
  5. We have rewritten much of Section 4 (Conclusions/Summary) including additional thoughts on potential future therapies.
  6. A few miscellaneous changes have been made that we believe better explains some of the data and/or figures.

All changes appear in red print for easy access.

Again, we wish to thank the Reviewers for their excellent insights and suggestions for improving the overall manuscript.

Manuscript IJMS-2470620

Response to Reviewer 1 & Reviewer 2

We thank both Reviewer 1 and Reviewer 2 for taking time to review manuscript ijms-2470620 that was submitted to IJMS as a perspective report comparing the early pathological events occurring in the lacrimal glands versus the salivary glands primarily of the SS-susceptible C57BL/6.NOD-Aec1Aec2 mouse model.  Although several studies have been published describing both early and later events in the lacrimal glands and in the salivary glands of this mouse model, we have not directly compared these two data sets per se.  Thus, the current manuscript.

Both Reviewer 1 and Reviewer 2 identified a few weaknesses that needed to be addressed.  At this time, we have addressed these points, plus a few additional ones,  and feel that addressing these issues have greatly improved the overall paper. More specifically:

  1. The Title of the manuscript has been changed significantly to better describe the purpose of the overall paper.
  2. We have rewritten the title for Section 2.2 of the Results section.
  3. We have modified the figure legend for Figure 4.
  4. While a perspective paper does not require a methods section, we have included a statement in the Conclusion/Summary section mentioning where the data are available for viewing … in addition to the Data Availability Statement.
  5. We have rewritten much of Section 4 (Conclusions/Summary) including additional thoughts on potential future therapies.
  6. A few miscellaneous changes have been made that we believe better explains some of the data and/or figures.

All changes appear in red print for easy access.

Again, we wish to thank the Reviewers for their excellent insights and suggestions for improving the overall manuscript.

Manuscript IJMS-2470620

Response to Reviewer 1 & Reviewer 2

We thank both Reviewer 1 and Reviewer 2 for taking time to review manuscript ijms-2470620 that was submitted to IJMS as a perspective report comparing the early pathological events occurring in the lacrimal glands versus the salivary glands primarily of the SS-susceptible C57BL/6.NOD-Aec1Aec2 mouse model.  Although several studies have been published describing both early and later events in the lacrimal glands and in the salivary glands of this mouse model, we have not directly compared these two data sets per se.  Thus, the current manuscript.

Both Reviewer 1 and Reviewer 2 identified a few weaknesses that needed to be addressed.  At this time, we have addressed these points, plus a few additional ones,  and feel that addressing these issues have greatly improved the overall paper. More specifically:

  1. The Title of the manuscript has been changed significantly to better describe the purpose of the overall paper.
  2. We have rewritten the title for Section 2.2 of the Results section.
  3. We have modified the figure legend for Figure 4.
  4. While a perspective paper does not require a methods section, we have included a statement in the Conclusion/Summary section mentioning where the data are available for viewing … in addition to the Data Availability Statement.
  5. We have rewritten much of Section 4 (Conclusions/Summary) including additional thoughts on potential future therapies.
  6. A few miscellaneous changes have been made that we believe better explains some of the data and/or figures.

All changes appear in red print for easy access.

Again, we wish to thank the Reviewers for their excellent insights and suggestions for improving the overall manuscript.

Reviewer 2 Report (New Reviewer)

In this perspective/ review, the Authors present data from previous publications (several by the Authors themselves)  as a basis for explaining the initially different pathological events that drive SS in lacrimal and salivary glands, and the subsequently similar upregulation of the TLR4/TLR3 signal transduction pathway that activates the type-1 interferon signature.

The manuscript is well written overall with pertinent use of data from murine models of SS to support their observations and concept. However, the title of the manuscript does not seem to reflect the content of the manuscript and I would strongly suggest changing it to better reflect the manuscript content, which does not focus on dacryoadenitis. 

In the abstract the Authors state that they will discuss the therapeutic implications of the findings they discuss but this was lacking. They should include more information on how they think these findings would impact therapeutic decisions.  

Please correct the typo "withrequiresin" in the legend of Figure 2.

The resolution of the figures and Table could be improved. 

Author Response

Manuscript IJMS-2470620

Response to Reviewer 1 & Reviewer 2

We thank both Reviewer 1 and Reviewer 2 for taking time to review manuscript ijms-2470620 that was submitted to IJMS as a perspective report comparing the early pathological events occurring in the lacrimal glands versus the salivary glands primarily of the SS-susceptible C57BL/6.NOD-Aec1Aec2 mouse model.  Although several studies have been published describing both early and later events in the lacrimal glands and in the salivary glands of this mouse model, we have not directly compared these two data sets per se.  Thus, the current manuscript.

Both Reviewer 1 and Reviewer 2 identified a few weaknesses that needed to be addressed.  At this time, we have addressed these points, plus a few additional ones,  and feel that addressing these issues have greatly improved the overall paper. More specifically:

  1. The Title of the manuscript has been changed significantly to better describe the purpose of the overall paper.
  2. We have rewritten the title for Section 2.2 of the Results section.
  3. We have modified the figure legend for Figure 4.
  4. While a perspective paper does not require a methods section, we have included a statement in the Conclusion/Summary section mentioning where the data are available for viewing … in addition to the Data Availability Statement.
  5. We have rewritten much of Section 4 (Conclusions/Summary) including additional thoughts on potential future therapies.
  6. A few miscellaneous changes have been made that we believe better explains some of the data and/or figures.

All changes appear in red print for easy access.

Again, we wish to thank the Reviewers for their excellent insights and suggestions for improving the overall manuscript.

This manuscript is a resubmission of an earlier submission. The following is a list of the peer review reports and author responses from that submission.

Round 1

Reviewer 1 Report

This is an excellent and comprehensive review of SS pathogenesis. I recommend to publish it.

Reviewer 2 Report

The paper is a hybrid review research article presenting both data and addressing published findings. The publication is presented as both a research article and a review, however, it should be reformatted as a review and original claims deleted. 

The presentation of the data from unpublished and published studies is erratic. The data presented without citations lack credibility, there are no statistics, qPCR or westerns to verify these claims at scale. The data presented with citations often lack independent reproducibility. Most of the data presented focus on one set of data collected in 2009 which form the basis for a large number of subsequent publications. This is problematic because most lack low throughput reproduction. It is not clear the data analysis techniques used are valid, and there is little discussion of the extent of the changes, the fold changes described in those manuscripts are relatively small. 

e.g.

MZB cells differentiate from transitional type-1 (T1) B cells under the influence of low affinity B cell receptor (BCR) signals and transcription factors, especially Notch-2 [55-58]. Activation of MZB cells involves the stimulation of their BCR complex by deltalike ligands or Hedgehog ligands that, in turn, activates the Notch2 signal transduction pathway that, in the abence of Notch1, Notch3 or Notch4 co-activation, is unique to the MZB cells. As shown in Figure 3, a common Notch2 signal transduction pathway appears to be present in the lacrimal and salivary glands despite the fact that each uses different Jag (Jag1 versus Jag2) and Dtx (Dtx4 versus Dtx1) molecules (Citation!!!!!). However, the Notch2 signal transduction profile(s) exhibited in the salivary glands suggests a possible second pathway using the receptor Mfng and several unique signaling molecules (Citation!!!!).

Several unsupported claims are made:

e.g.

2.4 The upregulated gene expression profiles for the interferon type-1 signal transduction pathways expressed in salivary and lacrimal glands of C57BL/6.NOD-Aec1Aec2 mice are not only similar gene sets, but expressed during an identicaltime frame. 

But there is no proof of this with corresponding data or a figure and no citation proving this citation. 

Figure 2. A partial overview of major temporal changes identified in the progressive development of Sjögren´s Syndrome pathology. Based on physiological and histological changes and the various time points for the appearance of specific immune cell populations within the lacrimal and salivary glands, it is possible to divide the overall disease into three main compartments. Importantly, these general disease profiles occur at the same time periods in the lacrimal and salivary glands (cite). In addition, the onset of lymphomagenesis, primarily B cell lymphomas, appears to be a function of interleukin IL14α (alpha-taxilin) expression (cite), in that the IL14α gene in C57BL/6.NOD-Aec1Aec2 mice, which is upregulated in both lacrimal and salivary glands during the early innate immunity phase (cite), is quickly down-regulated following the transition to the adaptive immune phase (data not presented) - But this is review so cite or remove . These data strongly suggest that lymphomagenesis in SS requires continued IL14α stimulation. (But how do we know that) Also note that these physiological changes are highly transitional across phases once activated (But why?)

Citation 46 was the wrong paper for the image, please ensure copyright clearance. Is this actually essential to understanding the paper. 

Reviewer 3 Report

The manuscript entitled "Are Marginal Zone B Cells the Basis for Dacryoadenitis in Experimental Sjögren´s Syndrome" is a good work done by the authors. However, there are some clarifications are needs in some areas before further consideration. Please find my comments and suggestions below;

1. The abstract must state the background of the study more clearly (including the significance of the Sjögren´s Syndrome)

2. The status and prevalence of the Sjögren´s Syndrome is not mentioned clearly in the introduction. It needs to be included.

3. Citation of references in introduction is found to be like "[1-6],  [14-17]". I suggest to split these into individual references and explain each. You can combine the concept of two or three article, not more than that.

4. Cytokine profile of SS patients are important in the immune disorder. However, there is no mention about it.

5. In the results, figure 1: there is no description on the magnification of the specimen. It will also be beneficial to include a scale in the image

6. "This figure is reproduced from [46]" line incomplete 

7. "with an upregulated Tlr3 expression still present at 20 weeks of age (data not presented)" It will be better to include the data as supplementary file